# Fast and pervasive diagenetic isotope exchange in foraminifera tests is species-dependent

Deyanira Cisneros-Lazaro [1✉], Arthur Adams [1], Jinming Guo [1], Sylvain Bernard[2], Lukas P. Baumgartner[3], Damien Daval[4], Alain Baronnet[5], Olivier Grauby[5], Torsten Vennemann[6], Jarosław Stolarski [7], Stéphane Escrig[1] & Anders Meibom [1,3✉]

Oxygen isotope compositions of fossil foraminifera tests are commonly used proxies for ocean paleotemperatures, with reconstructions spanning the last 112 million years. However, the isotopic composition of these calcitic tests can be substantially altered during diagenesis without discernible textural changes. Here, we investigate fluid-mediated isotopic exchange in pristine tests of three modern benthic foraminifera species (*Ammonia sp.*, *Haynesina germanica*, and *Amphistegina lessonii*) following immersion into an $^{18}$O-enriched artificial seawater at 90 °C for hours to days. Reacted tests remain texturally pristine but their bulk oxygen isotope compositions reveal rapid and species-dependent isotopic exchange with the water. NanoSIMS imaging reveals the 3-dimensional intra-test distributions of $^{18}$O-enrichment that correlates with test ultra-structure and associated organic matter. Image analysis is used to quantify species level differences in test ultrastructure, which explains the observed species-dependent rates of isotopic exchange. Consequently, even tests considered texturally pristine for paleo-climatic reconstruction purposes may have experienced substantial isotopic exchange; critical paleo-temperature record re-examination is warranted.

[1] Laboratory for Biological Geochemistry, School of Architecture, Civil and Environmental engineering, Ecole Polytechnique Fédérale de Lausanne (EPFL), CH-1015 Lausanne, Switzerland. [2] Museum National d'Histoire Naturelle, Sorbonne Université, CNRS UMR 7590, IMPMC, 75005 Paris, France. [3] Center for Advanced Surface Analysis, Institute of Earth Science, University of Lausanne, CH-1015 Lausanne, Switzerland. [4] ISTerre, Univ. Grenoble Alpes, Univ. Savoie Mont Blanc, CNRS, IRD, IFSTTAR, 38058 Grenoble, France. [5] CNRS, CINaM, Aix-Marseille Université, 13009 Marseille, France. [6] Institute of Earth Surface Dynamics, University of Lausanne, CH-1015 Lausanne, Switzerland. [7] Institute of Paleobiology, Polish Academy of Sciences, PL-00-818 Warsaw, Poland. ✉email: deyanira.cisneroslazaro@epfl.ch; anders.meibom@epfl.ch

**B**iogenic calcite chemical and isotopic compositions offer scientists a broad range of proxies for past environmental conditions[1]. Arguably the most important among these proxies is the oxygen isotope composition, which was developed as a paleothermometer by Urey and colleagues more than six decades ago[2–5]. Since then, the oxygen isotopic composition of carbonate has been systematically used to reconstruct past ocean temperatures. In particular, the oxygen isotope composition of fossil calcitic tests of benthic foraminifera has produced a reconstruction of the paleotemperature of the deep ocean with high temporal resolution as far back as 112 Myr[6].

The oxygen isotopic composition of biogenic carbonate reflects both the oxygen isotope composition and the temperature of the seawater in which the biomineralizing organisms live, as well as the biological mechanisms involved in their biomineralization process (so-called vital effects)[3,7–10]. The use of the oxygen isotope paleothermometer relies on the widely accepted paradigm that fossil foraminifera tests collected from ocean sediments have retained their original elemental and isotopic compositions; at least those tests that appear pristine when observed with optical microscopy or scanning electron microscopy (SEM)[11–16]. However, from the moment a marine organism dies, diagenetic processes may begin to alter the original isotopic and chemical composition of its biocarbonate remains[17] and might thus introduce bias in subsequent paleo-environmental reconstructions.

Calcium carbonate skeletons, shells, and tests formed by marine animals have a nanocomposite organo-mineral structure[18]. The tests of planktonic and benthic foraminifera—the latter being the focus of this study—are made up of irregular (sub-spherical) nanocrystallites of calcite 10–100 nm in diameter[18–20]. Furthermore, these tests are compositionally heterogenous at the mesoscale, incorporating elements, such as Mg, Na, P, and S[21–23]. The banded distributions of these elements reflect, to different degrees, the presence and distribution of organic matter within the test wall[22,24–26]. This ultrastructure creates a very large internal surface on which elemental and isotopic exchange with sediment porewater can take place, provided that this water can penetrate deep enough into the structure, which seems to be the case according to several recent studies[27,28].

Bernard et al.[27] subjected planktonic foraminifera tests to elevated temperature and pressure (300 °C, 200 bar) in pure $H_2^{18}O$ artificial seawater at chemical equilibrium with calcite and demonstrated that, within months, the tests had exchanged up to 15 vol% of their $^{18}O$ with the seawater analogue without observable (by SEM) changes to test ultrastructure and morphology. Assuming solid-state diffusion as the dominant process, these authors showed that substantial bulk oxygen isotopic exchange of a fossil foraminifera tests can take place on a timescale of 10 Myr under ambient conditions in ocean sediments, resulting in paleotemperature overestimations and explaining the unrealistically flat inferred temperature gradients in the paleo-ocean as a result of diagenetic bias.

The rapid isotopic exchange between carbonate and an aqueous phase has subsequently been experimentally confirmed for a series of carbonate minerals at much lower temperatures (down to room temperature), on time scales never exceeding a few months. Chanda et al.[28] used $^{45}Ca$ as a radiotracer to investigate the recrystallization of planktonic foraminifera tests in seawater analogues in equilibrium with calcite at 25 °C. The radiotracer was incorporated into the biocalcite within days with minimal structural change and it was concluded that intra-test chemical heterogeneities played a key role in the recrystallization of these tests. Other experiments with micrometer-sized abiotic carbonates—including calcite—in chemical equilibrium with surrounding fluids have demonstrated that isotopic exchange can proceed at room temperature even in the absence of chemical heterogeneities[29–32].

In this study, we expose three species of foraminifera to precisely controlled experimental conditions simulating diagenesis to evaluate how biogenic calcites interact with surrounding fluids. A combination of NanoSIMS imaging and scanning- and transmission electron microscopy (SEM and TEM) investigations permits to visualize and quantify isotope exchange inside the experimentally fossilized calcitic tests while documenting relevant ultrastructural features. We show that isotopic exchange rates vary consistently across the three species studied, and we relate this to species-specific differences in ultrastructure. Together, these observations provide insight into the processes that drive isotopic exchange in fossil foraminifera tests and, by analogy, other fossil biocalcites.

## Results

**Oxygen isotope exchange experiments**. Initially pristine tests of three species of recently collected modern benthic foraminifera, *Ammonia* sp., *H. germanica*, and *A. lessonii*, as well as abiotic Iceland spar calcite crystals, were incubated for 6 days in a highly $^{18}O$-enriched ($^{18}O/^{16}O = 0.30$) seawater analogue in chemical equilibrium with calcite at 90 °C; additional experiments lasting only 4 h were conducted on *Ammonia* sp. and *A. lessonii* tests under the same conditions. Before these experiments, the tests were treated with a standard methanol and oxidative cleaning procedure, which removes the organic cell materials (mostly proteins and polysaccharides) as well as a large fraction of inter-crystalline organic materials[33]. This procedure mimics the natural fossilization process, in which the majority of organic material is rapidly degraded upon the death of the organism[33–37]. Bulk measurements of the O-isotope composition (i.e., the $^{18}O/^{16}O$ ratio) of experimentally fossilized tests were performed to precisely quantify isotopic exchange.

**Ultrastructure of the starting materials**. The spiraled, multi-chambered, calcitic tests of the pristine benthic species used in this study are similar to the tests of benthic foraminifera species most frequently used for paleo-environmental reconstructions. In their natural pristine state, they are semi-transparent (i.e., hyaline or glassy) and perforated by pores (hence the name foraminifera, which in Latin means hole bearer, Figs. 1 and 2d, f–h). At the formation of each new test chamber, a calcitic layer is formed on top of pre-existing chambers, separated by a layer of organic matter; the so-called organic linings[21,22,25,26,38,39]. This creates a test wall consisting of calcite-dominated layers separated by relatively thin (100–300 nm) sheets of organic linings oriented parallel to the wall surface[22,40,41]; these organic linings and the calcite layers they separate are clearly seen in fractured test walls (Fig. 2b). The test wall is further divided into domains that resemble interlocking cogwheels when observed on the outer wall surface, or in the interior of the wall in sections cut parallel to the wall surface (Fig. 2d–i)[42,43]. The cogwheel-like domains are 3-dimensional structures that extend throughout the thickness of the wall (Fig. 2c). Each of these cogwheel-like structures consists of 10–100 nm sub-spherical calcite-dominated particles sharing a common crystal orientation, which makes each cogwheel domain appear as a single crystal in bright-field TEM images (Fig. 2e, i). The interfaces between individual cogwheel structures, which are also enriched in organic matter[44], are readily imaged with SEM (Fig. 2d, f, h, g). Therefore, in addition to the organic linings, the test wall also contains organic-rich interfaces separating individual cogwheel domains, albeit with the latter oriented perpendicular to the surface of the test wall; i.e., parallel to the local pore axis (Fig. 2c). These key features are illustrated schematically in Fig. 2a. Importantly, the size and organization of cogwheel structures, as well as the density and tortuosity of their interfaces,

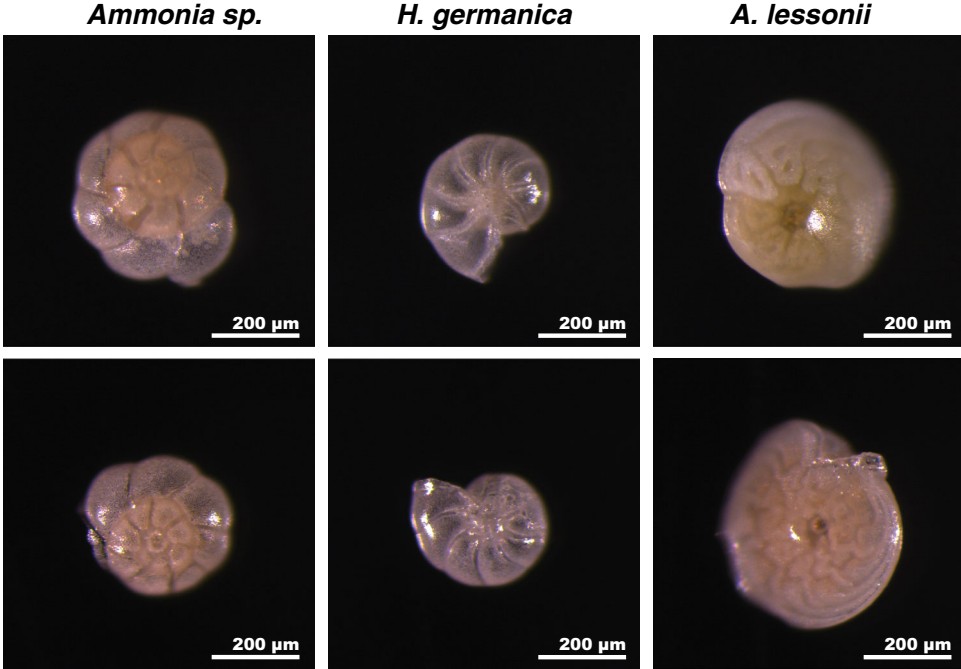

**Fig. 1 Appearance of tests before and after incubation.** Stereo microscopy images of *Ammonia* sp., *H. germanica*, and *A. lessonii* tests before (top row) and after (bottom row) incubation for 6 days at 90 °C in artificial seawater (ASW) with a $^{18}O/^{16}O$ ratio of 0.30. All images were taken at the same scale, under identical illumination conditions, and using the same microscope and camera.

vary from one species of foraminifera to another, as Fig. 2 illustrates. Finally, at the nanoscale, the 10–100 nm sub-spherical calcite-dominated particles (Supplementary Fig. 1) have been shown through atomic force microscopy (AFM) to be separated by organic-rich grain boundaries[18]. Due to the cleaning procedures (cf. "Methods" section) a large fraction of these organic materials were broken down and partly removed before exposure to the experimental fossilization conditions.

**Ultrastructure of the experimentally incubated materials**. The first and fundamentally important observation from the 6-days-long experiments at 90 °C was that it was practically impossible to distinguish between pristine tests (i.e., the starting materials) and tests having been exposed to diagenetic conditions (i.e., the experimentally incubated materials) (Fig. 1, Supplementary Figs. 1–7). Both the pristine and experimentally incubated tests appeared translucent and glassy, with chamber divisions and pores easily recognizable using reflected light and a stereo-microscope. Even at the level of SEM imaging, it is not possible to distinguish between uncleaned foraminifera tests and tests that were exposed to the experimental solution (Supplementary Figs. 1, 5–7).

**Species-specific bulk $^{18}O$-enrichments**. Bulk oxygen isotope measurements were made on calcite tests that have been exposed to the experimental solution for 6 days, with each batch containing 70–100 µg of foraminifera. The bulk $^{18}O$-enrichments reported in parts-per-thousand relative to VSMOW ± 1 standard deviation (SD) were 529 ± 49‰ ($n = 10$) for *Ammonia* sp., 769 ± 80‰ ($n = 10$) for *H. germanica*, and 709 ± 36‰ ($n = 12$) for *A. lessonii* (Fig. 3, Supplementary Table 1). To determine whether the bulk $^{18}O$-enrichments per species were significantly different ($p < 0.05$) from one another, unpaired two-tailed *t*-tests were made after testing for variance with F-tests. *Ammonia* sp. bulk $^{18}O$-enrichments were

significantly different, $p = 9 \times 10^{-9}$ and $p = 5 \times 10^{-7}$, between *H. germanica* and *A. lessonii*, respectively, but those between *H. germanica* and *A. lessonii* ($p = 0.06$) were not. Overall, these results demonstrate that substantial O-isotopic exchange occurred between the seawater analogue and the calcitic tests during the experiments and that the degree of isotopic exchange is species-specific. Note that organic compounds remaining within the tests should not significantly contribute to these measured values[45,46].

**NanoSIMS imaging of $^{18}O$-enrichments**. The $^{18}O/^{16}O$ ratio variations in the tests exposed to the experimental solutions for 6 days were imaged with a NanoSIMS ion microprobe[47], on surfaces both perpendicular and parallel to the local axis of pores, revealing localized $^{18}O$-enrichments that explain the bulk measurements. Note that the $^{18}O$-enrichments measured with the NanoSIMS are reported in parts-per-thousand relative to a foraminifera test with natural isotopic composition and that the measured $^{18}O/^{16}O$ ratios are completely dominated by secondary ions of O⁻ derived from calcite. The secondary ion yield of O⁻ differs for calcite and organics, but neither in NanoSIMS images nor in line-scans were changes in the $^{16}O^-$ count rate across the test observed to correlate with regions enriched in P, which point to remains of organic linings[22]. In *Ammonia* sp., surfaces parallel to the pore axis (i.e., perpendicular to the test surface) show thin (~0.3 µm) bands of $^{18}O$-enrichment that followed the curvature of the test wall (Fig. 4a). These $^{18}O$-enriched bands correlated systematically with P-enrichments (Fig. 4a, d) indicating remains of organic linings that had withstood the oxidative cleaning procedure[22]. Consistently, the number of $^{18}O$-enrichment bands correspond to the number of organic linings expected from the specific location of the section within the test and the total number of chambers (Supplementary Fig. 8). All three investigated species thus had qualitatively similar correlation between P-enrichments and $^{18}O$-enrichment bands (Fig. 4). However, it is noted that

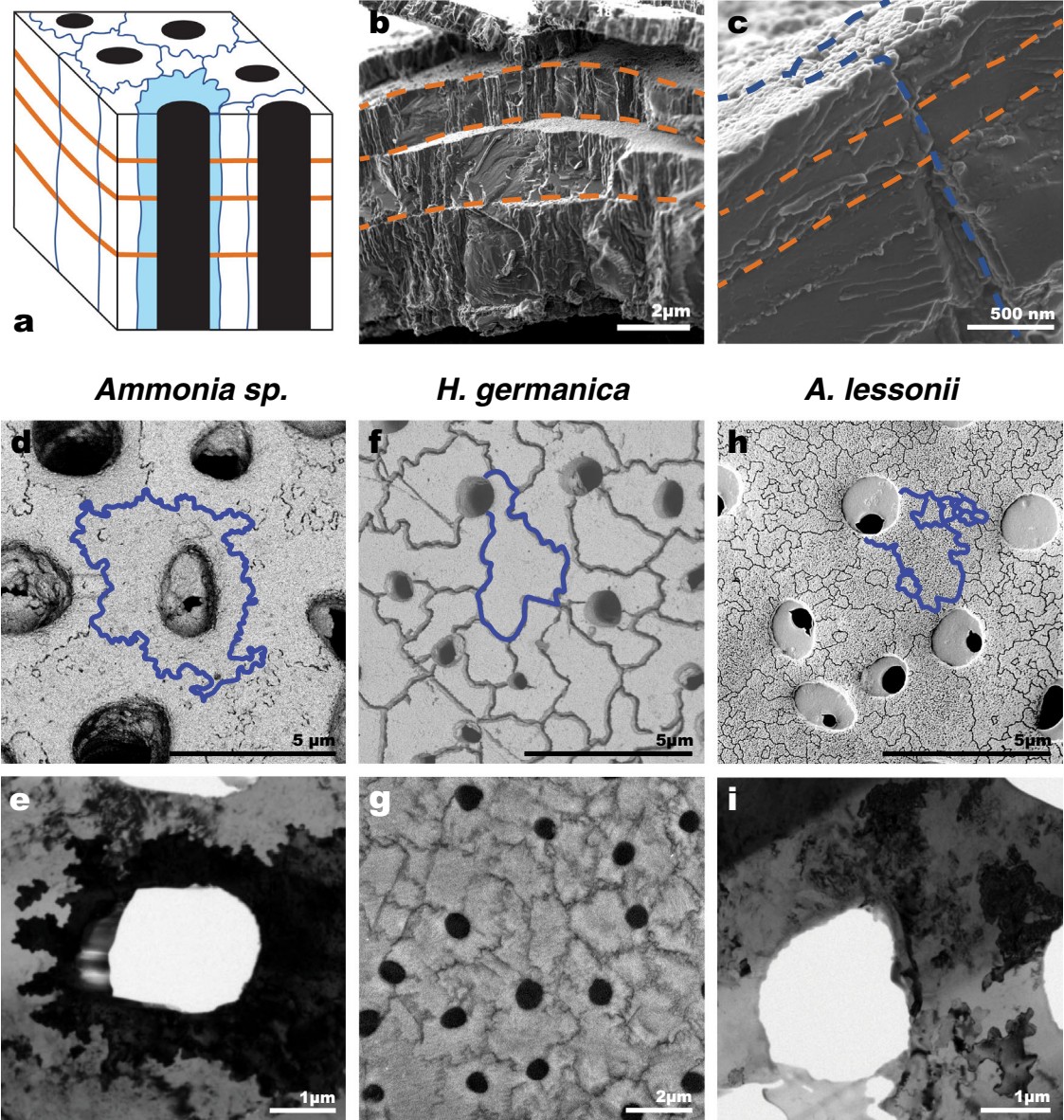

**Fig. 2 Ultrastructural organization.** Organic linings and cogwheel structures in *Ammonia* sp., *H. germanica* and *A. lessonii*. **a** Schematic diagram showing the spatial location of cogwheel structures and organic linings on the surface and inside benthic foraminifera test walls. Cogwheel interfaces are outlined in blue, a single cogwheel structure is shaded in light blue, pore-spaces are black, and organic linings are marked by orange lines. **b** SEM image of the edge of a broken *Ammonia* sp. test wall. The step-like fractures running parallel to the test surface (highlighted in orange hatched lines) correspond to the organic linings. **c** SEM image of the surface and interior of an *Ammonia* sp. test wall showing a cogwheel interface (outlined with dashed blue lines) extending throughout the thickness of the wall, perpendicularly to the organic linings (dashed orange lines). **d** SEM image of cogwheels on the surface of an *Ammonia* sp. test. **e** Bright-field (BF) TEM image centred on a single cogwheel in a FIB thin-section obtained from the interior of an *Ammonia* sp. test wall (cut parallel to the wall surface). The dark cogwheel in the centre has a slightly different crystallographic orientation relative to surrounding cogwheels (light grey). **f** SEM image of the surface of *H. germanica* showing large, irregularly shaped cogwheels. **g** SEM image of a lightly EDTA etched section from the interior of a *H. germanica* test, which is parallel to the wall surface, demonstrating that cogwheel structures are continuous throughout the test walls. **h** SEM image of the surface of a lightly EDTA etched *A. lessonii* test exhibiting cogwheels with variable surface areas. **i** BF TEM image of a FIB thin section obtained from the interior of an *A. lessonii* test wall (cut parallel to the test surface). The small cogwheels have jig-saw like interfaces between them.

within most imaged *Ammonia* sp. tests ($n = 18$) there was one additional band of [18]O-enrichment close to the inner edge of the shell not associated with a band of P-enrichment (Fig. 4a, d). Perpendicular to these thin P- and [18]O-enriched bands were broader, more irregular zones of [18]O-enrichments of varying thickness running across the full width of the test (Fig. 4a). On surfaces oriented perpendicular to the local pore axis, i.e., parallel to the test surface, the [18]O-enrichments in *Ammonia* sp. formed tortuous line patterns around the pores, identical in appearance to

those outlined by the interfaces between cogwheel domains (compare Fig. 4g and 2d). Qualitatively similar [18]O-enrichment patterns were observed in *H. germanica* and *A. lessonii* tests experimentally incubated under identical conditions (i.e., 6 days at 90 °C) (Fig. 4), and these [18]O-enrichments directly reflect the species-specific differences in density and tortuosity of the cogwheel domain interfaces (Figs. 2 and 4).

Furthermore, NanoSIMS line-scans were made parallel to the pore axis in tests of *Ammonia* sp. and *H. germanica* to

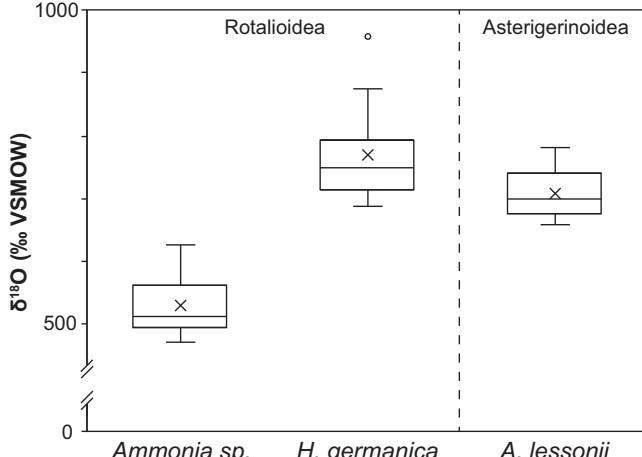

**Fig. 3 Bulk $^{18}$O-enrichment after incubation.** Box plot distributions of bulk $^{18}$O-enrichment in tests of the three benthic foraminifera species incubated for 6 days at 90 °C in seawater analogue with a $^{18}$O/$^{16}$O ratio of 0.30. The average $^{18}$O-enrichments are shown with crosses. The superfamily that the foraminifera species belong to are indicated at the top of the figure. Middle lines = medians, boxes = interquartile ranges, whiskers = minimum and maximum values, empty circle = outlier.

investigate the level of $^{18}$O-enrichment in test material in-between organic linings and cogwheel interfaces (Fig. 5). These line-scans showed that the $^{18}$O-enrichments reached 1500‰ where P concentrations indicated remains of organic linings. In addition, the adjacent test calcite (i.e., in-between organic linings) was also clearly enriched by about 100‰ above the $^{18}$O/$^{16}$O value of an isotopically unlabelled control sample measured under identical conditions.

Taken together, these observations indicate that isotopic exchange occurred between the foraminifera calcite particles and the $^{18}$O-enriched seawater analogue, not only along the organic linings and cogwheel interfaces, but also throughout the test, albeit to a lesser extent. A series of short 4-h experiments with *Ammonia* sp. and *A. lessonii* tests were conducted under the same conditions and provided qualitatively similar results, but with correspondingly lower $^{18}$O-enrichments (Supplementary Fig. 9). This demonstrated that isotopic exchange took place on a timescale of minutes and hours.

**The difference in reactivity between biogenic and abiotic calcite.** The exact same 6-day-long experiment was conducted with cleaved, mm-sized single crystals of Iceland spar yielding strikingly different results (Supplementary Fig. 10). No $^{18}$O-enrichments were measured in NanoSIMS images in the interior of these Iceland spar crystals analysed under identical conditions and at the same scale as the foraminifera tests (Supplementary Fig. 10). This underlines the fundamental difference in reactivity between a biogenic calcite, here in the form of foraminiferal tests with their intricate organo-calcite composite internal structures (e.g., linings and cogwheel interfaces), and a purely abiotic calcite crystal.

## Discussion
**Localized $^{18}$O-enrichments.** NanoSIMS images and line-scans on surfaces oriented parallel to the local pore axis show that the bands of high $^{18}$O-enrichment follow the curvature of the test wall, and are correlated to bands rich in P (Figs. 4 and 5). Phosphorous enriched bands in foraminifera tests are exclusively associated with organic linings and the P-enriched bands

measured here point to remains of organic linings having withstood oxidative cleaning[22]. During the life of foraminifera, these organic linings play a key role in directing calcite nucleation during the process of foraminifera test chamber formation[48–52]. Indeed, the number of $^{18}$O-enriched bands found in the test wall at specific positions matches the number of organic linings expected at these positions within the tests (Supplementary Fig. 8).

At the location of these $^{18}$O-enrichment bands, there are often (but not always) also bands with higher concentrations of S and Mg (Supplementary Fig. 8). Sulfur can be hosted by organic molecules or incorporated within the calcite lattice as $SO_4^{2-}$ [53]. Magnesium can substitute for Ca in the calcite lattice[54], but is also concentrated within the organic linings[41]. Chanda et al.[28] argued that stable mineral recrystallization of foraminifera tests proceeds through the preferential dissolution of Mg- and Sr-rich calcite by a dissolution-precipitation mechanism. However, our NanoSIMS images indicate that Mg- and S-rich bands are broader and more numerous than the $^{18}$O-enrichment bands, and the tests do not show evidence of any significant dissolution or precipitation (Supplementary Figs. 5–7).

The close association between $^{18}$O-enrichment and P-rich bands demonstrates that organic linings are sites of preferential isotopic exchange as $H_2O$ penetrates into the spaces created by the partial breakdown of the organic molecules caused by oxidative cleaning. This is analogous to what will occur postmortem inside a foraminifera test in natural settings, where the intercrystalline organic matter will be (at least partially) degraded[33–37], paving the way for water penetration and subsequent isotopic exchange.

On surfaces perpendicular to the local pore axis (i.e., parallel to the surface), the tortuous lines of $^{18}$O-enrichment in all three species (Fig. 4g–i) are clearly correlated to the shape, distribution, and density of the cogwheel structures and their interfaces (Fig. 2d, f, h). Consistently, on surfaces parallel to the local pore axis, the cogwheel interfaces were apparent as undulating sheet-like zones of $^{18}$O-enrichments (Fig. 4a–c). Interfaces between cogwheel structures initially contain abundant organic matter and therefore, by analogy with the organic linings, represent another site for preferential $H_2O$ penetration and isotopic exchange; $H_2O$ penetrates into the spaces created by the partial breakdown of the organic molecules during oxidative cleaning and by extension, during diagenesis.

**Pervasive $^{18}$O-enrichments.** Partially broken-down organic linings and organic materials at cogwheel interfaces provide highways for $H_2O$ penetration and efficient isotopic exchange with adjacent calcite, and therefore exhibit the highest $^{18}$O-enrichments. However, line-scans across *Ammonia* sp. and *H. germanica* (Fig. 5) revealed $^{18}$O-enrichments averaging 100‰ (compared to unlabelled test calcite) in regions between organic linings and without apparent cogwheel interfaces, demonstrating that isotopic exchange also took place elsewhere in the test, at least down to a length-scale below the resolution of the NanoSIMS instrument (i.e., about 100 nm).

At the ultrastructural level, away from linings and cogwheel interfaces, organic matter is concentrated[18] between the 10–100 nm sub-spherical calcite particles that make up the bulk of the foraminifera tests (Supplementary Fig. 1). Partial breakdown of this organic matter will, by analogy with the organic linings and organic compounds along with the cogwheel interfaces, also create pathways for $H_2O$ penetration and surfaces for isotopic exchange with calcite everywhere in the tests at <100 nm length scales. Using P as a proxy for the presence of remaining organic compounds after oxidative cleaning, we

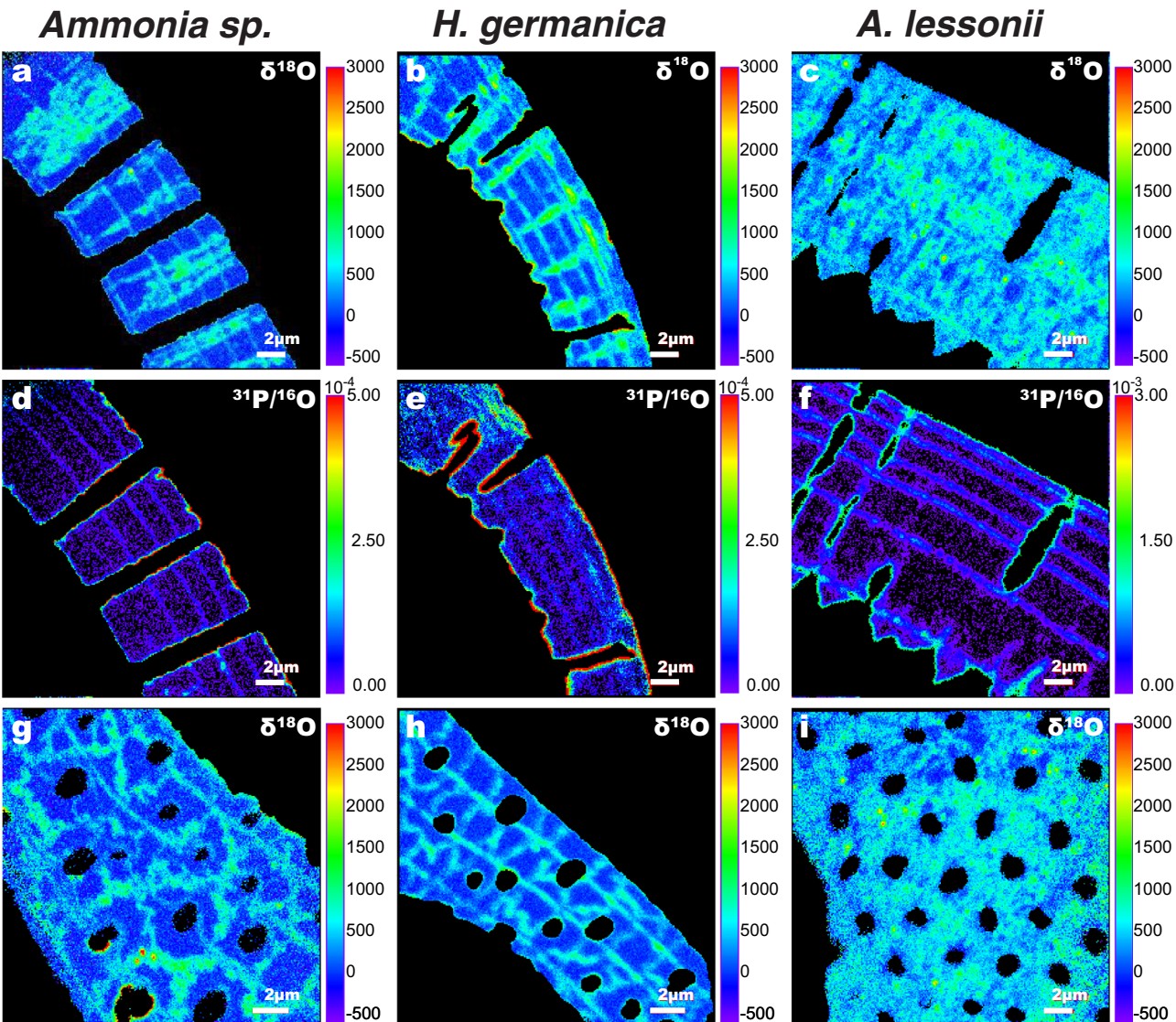

**Fig. 4 Direct imaging of ¹⁸O-enrichments in foraminifera tests after incubation.** NanoSIMS images of *Ammonia* sp., *H. germanica, and A. lessonii* tests after incubation for 6 days at 90 °C in seawater analogue with a ¹⁸O/¹⁶O ratio of 0.30. **a–c** Surfaces exposing the interior of test walls show high ¹⁸O-enrichments occurring as thin parallel bands following the curvature of the test walls and elongated zones of variable thickness running perpendicular to the test surface. **d–f** Same view of the same samples showing the ³¹P/¹⁶O ratio, which reveals P-rich bands following the curvature of the test walls. **g–i** Surfaces from the interior of the same samples, but cut parallel to the wall surface, show ¹⁸O-enrichments occurring as thin, tortuous and cogwheel-shaped lines in *Ammonia* sp. and *H. germanica*, which merge into patchy areas of higher ¹⁸O enrichment in *A. lessonii*.

observed that the P-counts did not decrease to zero anywhere in the tests (Fig. 5), indicating the presence of organic matter in regions between organic linings and away from cogwheel interfaces at length scales below the resolution of the NanoSIMS; a finding also reported by Geerken et al.[22]. At the atomic scale, the presence of these organic molecules causes anisotropic distortion of the crystal lattice of biogenic calcite[55]. Compared to abiotic calcite, biogenic calcite thus has significantly longer and weaker C–O bonds[56,57]. Consequently, where $H_2O$ preferentially penetrates the tests because of the partial breakdown of organic compounds, it also preferentially exchanges O-isotopes with the calcite surfaces it encounters.

The role of inter- and intra-crystalline organic matter in permitting water penetration and promoting isotopic exchange is further supported by the measured ¹⁸O-enrichments in the foraminifera tests exposed to the experimental solutions compared to O-isotope measurements in macroscopic (i.e., mm-sized) abiotic Iceland spar calcite exposed to the very same experimental

conditions (Supplementary Fig. 10). No ¹⁸O-enrichment was measurable in the interior of these Iceland spar crystals, in which water simply could not penetrate because of the lack of structural pathways that, in contrast, are ubiquitous in biogenic calcites in general[44,58–60].

**Species-specific susceptibility to isotopic exchange.** Organic linings and cogwheel structures are fundamental structural features in the tests of both benthic and planktonic foraminifera[43], and can become the main pathways for $H_2O$ penetration and hence also induce isotopic exchange after the (partial) degradation of the organic matter (Figs. 3–5). In addition to chamber walls of variable thickness, tests of different foraminifera species have different cogwheel structures in terms of size, distribution, and density[43], as is the case for the species studied here (Fig. 2). This would then also result in a species-specific susceptibility for the penetration of water and subsequent isotopic exchange. In

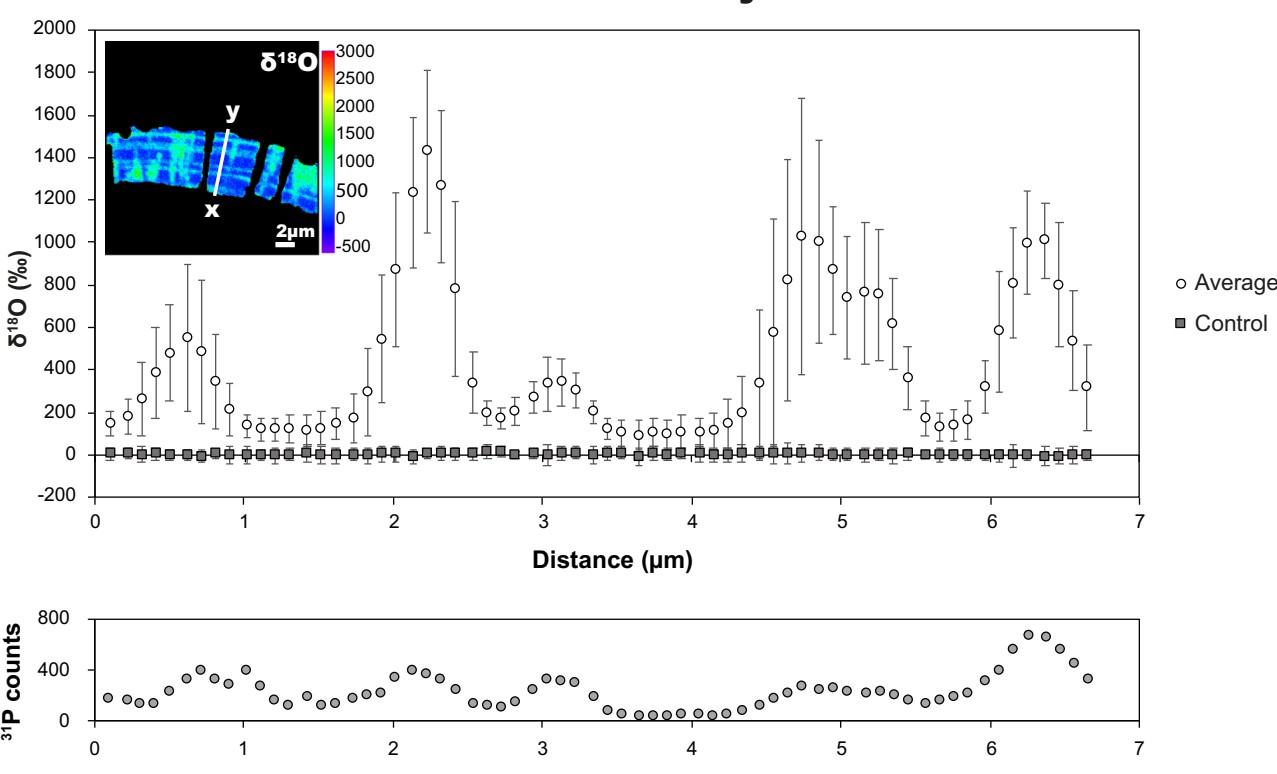

**Fig. 5 $^{18}$O-enrichments in test walls between organic linings and cogwheel interfaces.** Top: NanoSIMS line-scans across the width of a *H. germanica* test wall after incubation for 6 days at 90 °C in seawater analogue with a $^{18}$O/$^{16}$O ratio of 0.30 (white circles), compared to a control sample with natural $^{18}$O/$^{16}$O ratio (dark squares); error bars are 2$\sigma$. NanoSIMS image inset shows the location of the line going from *x* to *y*. Bottom: corresponding $^{31}$P-counts over the same line-scan indicating the location of organic linings.

other words, the different susceptibilities of biogenic carbonates to diagenesis appears directly correlated with differences in ultrastructure.

The cogwheel structures and the organic linings are the main sites for preferred isotopic exchange, therefore, a higher proportion of these structures in a foraminifera test supports a higher bulk $^{18}$O enrichment. The surface density of the cogwheel interfaces was calculated for all three species of foraminifera used in this study (Supplementary Fig. 11, Supplementary Table 2) using the methodology described in van Dijk et al.[43]. *Ammonia* sp. and *H. germanica* had comparable average cogwheel interface densities (measured in μm per μm$^2$) of $1.80 \pm 0.36$ ($1\sigma$) and $1.78 \pm 0.10$ ($1\sigma$), respectively, which were approximately half that of *A. lessonii*, $3.52 \pm 0.18$ ($1\sigma$). This higher cogwheel interface density of *A. lessonii* compared to *Ammonia* sp. is likely a contributing factor to this species having a higher average bulk $^{18}$O-enrichment than *Ammonia* sp.

Despite *Ammonia* sp. and *H. germanica* having similar cogwheel interface densities, the average bulk $^{18}$O-enrichment in *H. germanica* was 45% higher than in *Ammonia* sp. (Fig. 3, Supplementary Table 2). All three species belong to the same order, Rotaliida, but are differentiated at the Superfamily level, with *Ammonia* sp. and *H. germanica* belonging to Rotalioidea and *A. lessonii* to Asterigerinoidea. Consistent with this classification *Ammonia* sp. and *H. germanica* have similar test geometry (Supplementary Fig. 8, Supplementary Software), but *Ammonia* sp. test chambers appear thicker than in *H. germanica*. To quantify this, tests from both species were embedded in epoxy, ground down to the widest portion of the test and polished, imaged in SEM, and the images analysed using the image processing software ImageJ

(Supplementary Software). For both species, there is a good statistical correlation between the number of chambers and the cross-sectional test area measured in μm$^2$; *Ammonia* sp. $r^2 = 0.85$ and *H. germanica* $r^2 = 0.89$ (Fig. 6a). For the same number of chambers, *Ammonia* sp. tests occupy a larger area, indicating that test chambers are thicker than in *H. germanica*. Assuming these two species have organic linings of similar thicknesses (which is consistent with observations), the ratio of the cross-sectional test area to the number of chambers is both a proxy for chamber thickness and a measure of the density of calcite-organic interfaces. On average, *H. germanica* tests are 48% thinner than *Ammonia* sp. tests for the same number of calcite–organic interfaces (Supplementary Table 3), which is consistent with *H. germanica* having an average bulk $^{18}$O-enrichment that is 45% higher than in *Ammonia* sp. (Fig. 3, Supplementary Table 1).

The similar test geometry and cogwheel interface densities in *Ammonia* sp. and *H. germanica* offer the opportunity to study variations in diagenetic susceptibility to isotopic exchange during ontogeny. The ratio of the estimated maximum cross-sectional area of a test to its total number of chambers, which is equal to the number of organic linings separating consecutive layers in the test as it grows, was used to calculate a diagenesis resistance ratio. If this ratio is small, it indicates a relatively high number of organic linings for a given amount of calcite and therefore low resistance to isotopic exchange. Importantly, this ratio increases during ontogeny, i.e., with the number of completed chambers (Fig. 6b). From Fig. 6b, it is clear that *H. germanica* is relatively more susceptible to diagenesis than *Ammonia* sp. due to its thinner chambers. Additionally, smaller tests from younger foraminifera have fewer chambers and are

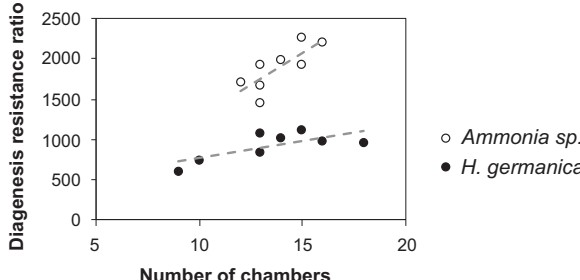

**Fig. 6 Variation in diagenetic susceptibility to isotopic exchange per species and during ontogeny.** Relationship between the number of chambers and foraminifera test cross-sectional area (**a**) and diagenesis resistance ratio for *Ammonia* sp. and *H. germanica*, calculated as a ratio of the foraminifera test cross-sectional area to the number of chambers (**b**). **a** The cross-sectional test wall area ($\mu m^2$) and the number of chambers show a good statistical correlation for both *Ammonia* sp. ($r^2 = 0.85$) and *H. germanica* ($r^2 = 0.89$). For the same number of chambers, *Ammonia* sp. test walls have larger cross-sectional areas than *H. germanica* tests because the walls are thicker in this species. **b** Plotting the diagenesis resistance ratio to the number of chambers shows that *H. germanica* has a lower resistance to diagenesis than *Ammonia* sp. and that, as foraminifera from both species grow more chambers, their diagenetic resistance ratio increases.

therefore more susceptible to diagenesis. This is because later added chambers are proportionally larger and their walls have fewer organic linings. Therefore, as foraminifera tests grow larger, their diagenetic resistance increases. This ontogenetic effect on diagenetic susceptibility can help explain some of the variability in the bulk isotopic enrichment measurements if foraminifera from a range of sizes is included in a single batch of analysed tests.

**Paleo-environmental reconstructions based on biogenic calcites—an outlook.** The effects of sediment lithology and sedimentation rate have been recognized as having an influence on foraminifera test preservation[61], with tests hosted in clay-rich sediments without significant breaks in sedimentation thought to be protected from porewaters with different chemical and isotopic compositions[11–13,61]. However, even these clay-rich oceanic sediments exhibit average porosities of 40% at burial depths up to 500 m[62], and thus contain abundant porewater. Even if the porewater surrounding the foraminifera tests maintain a constant chemical and isotopic composition, increased temperatures during the burial will lead to isotopic disequilibrium between the test calcite and the porewater, which drives isotopic exchange[27]. Therefore, consistent with the results of Bernard et al.[27] and Chanda et al.[28], our results make the case that partial isotopic exchange of foraminifera tests is unavoidable under natural conditions during diagenesis. We furthermore demonstrate that susceptibility to isotopic exchange is species-specific. Over millions of years, as carbonate skeletons approach isotopic equilibrium with the surrounding pore fluids in the sediments, this species-specific susceptibility to diagenesis will become less pronounced, which may partly explain the reduced variability of oxygen isotope ratios between different species over geological time; e.g., refs. [6,63]. It is common practice to correct compiled paleo O-isotopic records for assumed biological disequilibrium effects (the so-called vital effect)[3,63–66]. Species-specific susceptibility to diagenesis should also be corrected once its evolution over time is well established and quantifiable.

Furthermore, glassy fossil foraminifera tests should no longer be considered isotopically pristine. In modern paleo-environmental reconstruction studies, the collection of fossil foraminifera tests from ocean sediments for chemical or isotopic analysis involves a selection based on their appearance in optical microscopy: glassy tests are used in preference to frosty tests[12–16,67]. Frosty tests are

those that appear opaque when observed with an optical microscope because they have experienced extensive secondary crystallization; they are therefore generally discarded[13,17]. In contrast, glassy tests, i.e., those that appear transparent when observed with an optical microscope, are assumed to have undergone minimal diagenetic alteration in the form of neomorphism or cementation; i.e., replacement of original biogenic calcite with abiotic calcite or direct precipitation of calcite onto the tests[13]. But, as we demonstrate here, it is highly likely that glassy tests have also partially exchanged oxygen with a pore fluid. In fact, the benthic foraminifera tests exposed to the experimental incubations in this study can be considered to be texturally pristine, with no observable evidence of dissolution/precipitation or neomorphism. Yet, bulk measurements and quantitative Nano-SIMS imaging demonstrate that even though they appear visually pristine, these tests have undergone extensive isotopic exchange (Figs. 3–5). Additionally, since isotope exchange is ubiquitous throughout the entire tests, no amount of cleaning or fragmentation would remove their enriched isotopic compositions.

Moreover, even a cursory look at the literature shows that most (if not all) fossil foraminifera tests selected from geologically older ocean sediments (i.e., millions of years) and included in standard paleo-environmental reconstructions have been modified in terms of their initial texture, crystallinity, and ultrastructures during fossilization. Compare Fig. 1 and Supplementary Figs. 5–7 with Supplementary Fig. 12; the latter shows SEM images of fossil tests from the DSDP Site 522. Deeply etched pores and abundant secondary calcite cementation are features typical for the majority of tests from Paleogene ocean sediment sections[68] and this relatively poor level of preservation is certainly not the worst among fossil foraminifera tests for which the stable isotope compositions have been measured.

The results of the present study indicate that a texture-based assessment of the degree of diagenetic alteration of foraminifera tests is by itself not a sufficient criterium to exclude partial isotopic exchange after the sedimentation of the test. It also demonstrates that the effects of diagenetic isotopic exchange are species-specific. This needs to be considered and potentially corrected for in paleo-environmental reconstructions. In general, until diagenetic processes on different types of marine biocalcites are better understood and quantified, the existing paleo-seawater temperature reconstructions based on, for example, O-isotopic compositions cannot be considered unbiased.

## Methods

Three species of foraminifera, *Ammonia* sp., *A. lessonii* and *H. germanica*, were used in the autoclave experiments. *Ammonia* sp. and *H. germanica* were collected from recent sediment on tidal mudflats in the Bay of Bourgneuf, France. Two morphological features visible on the surfaces of tests, suture elevation and average pore diameter, were used to identify *Ammonia sp.* tests as phylotypes T1 and T6 as per the classification in Richirt et al.[69] *A. lessonii* were picked at depths of 15–45 m from the Gulf of Aqaba in Eilat, Israel.

**Test cleaning and oxygen isotope exchange experiments.** Before the experiments, foraminifera tests were cleaned following the standard methanol and oxidative cleaning procedures from Barker et al.[70]: clays adhered to foraminifera tests were removed through ultrasonication in methanol and deionized (MilliQ) water. Organic matter was removed by placing foraminifera tests in an alkali buffered 1% $H_2O_2$ solution in a boiling water bath for 20 min, followed by several rinses with deionized water and finally technical grade ethanol before overnight desiccation at 50 °C. Subsequently the three species of foraminifera and mm-sized cleaved single crystals of untreated Iceland spar were individually placed within flame-sealed glass ampules or welded gold capsules, hereby collectively referred to as autoclaves, along with ~100 μL of a seawater analogue ($\Omega_{calcite}$ = 1; 0.6 M NaCl, 0.05 M $MgCl_2$). The seawater analogue was enriched in $^{18}O$ to a $^{18}O/^{16}O$ ratio of about 0.30. Each autoclave was placed in an oven at 90 °C for 4 h or 6 days. Upon removal from the autoclaves, foraminifera and Iceland spar were rinsed in artificial seawater, distilled and deionized water, and technical grade ethanol and desiccated at 50 °C overnight, followed by 24 h of vacuum desiccation.

For bulk foraminifera analysis, 10–15 aliquots of 70 μg of dried and desiccated foraminifera per species (about 7–10 foraminifera) were analysed using a GasBench linked to a Finnigan Delta V (Thermo Fisher Scientific) mass spectrometer at the University of Lausanne according to a method adapted after Spötl and Vennemann[71]. Measured isotope ratios of desiccated foraminifera tests were identical to those of tests where the desiccation steps have been omitted, indicating that the measured isotope ratios are not influenced by any absorbed $^{18}O$-enriched water. Isotope ratios were normalized to the VSMOW scale using a Carrara marble in-house standard calibrated against NBS-19. Oxygen isotopic compositions are reported in per mil Vienna Standard Mean Ocean Water (VSMOW).

**NanoSIMS preparation and imaging.** For NanoSIMS analyses, the dried and vacuum desiccated samples were embedded in resin (EpoThin2, Struers) in aluminium rings. The resin was vacuum pumped to remove air bubbles and allowed to harden overnight. Sample surfaces were then polished using increasingly finer grained diamond paste (from 15 to 0.25 μm) producing a smooth cross-section of the chamber walls. The polished samples were coated with ca. 15 nm Au and the surface imaged with backscattered electrons using a Zeiss Gemini 500 SEM (University of Lausanne, UNIL) operating at an acceleration voltage of 20 kV and a working distance of 10–17 mm. NanoSIMS imaging of the resulting distribution of $^{18}O$-enrichment in the tests was carried out with a 16 keV $Cs^+$ primary ion beam focused to a spot size of about 120 nm (ca. 0.7 pA on the sample surface). Positive charge build-up on the surface was compensated by the use of an electron gun. The multi-collector system simultaneously counted the following ions in individual electron-multiplier detectors with a mass resolving power of ~9000 (Cameca definition): $^{16}O^-$, $^{18}O^-$, $^{28}Si^-$, $^{31}P^-$, $^{32}S^-$, $^{16}O^{24}Mg^-$, and $^{16}O^{40}Ca^-$. Areas of 25 × 25 μm or 30 × 30 μm were imaged with a raster of 256 × 256 pixels and a dwell-time of 5 ms per pixel. Up to 25 sequential images were produced for each area and were accumulated and drift corrected using L'IMAGE (developed by Dr. Larry Nittler, Carnegie Institution of Washington, USA). A threshold was applied to the images using the $^{16}O^{40}Ca^-$ counts, to only select for the calcitic test surface. $^{28}Si^-$ counts were used to eliminate rare clay particles adhering to the outer test surface. Oxygen isotope compositions are reported as $\delta^{18}O$ (in parts-per-thousand) relative to tests of pristine *Ammonia* sp., *A. lessonii*, *H. germanica*, or Iceland spar calcite that were not exposed to the experimental solutions:

$$\delta^{18}O = \left\{ \left[ (^{18}O/^{16}O)_{sample} - (^{18}O/^{16}O)_{standard} \right] / (^{18}O/^{16}O)_{standard} \right\} \times 1000 \quad (1)$$

**TEM preparation and imaging.** For the TEM analyses of *Ammonia* sp., test surfaces were precoated with a 10-nm-thick layer of carbon (for improved conduction and protection of the sample surface). Subsequent extraction and milling of lamellae that were parallel to the test surface were performed with a dual-beam Gemini NVision 40 Focused Ion Beam machine. The initial thick sections were milled with 30 kV Ga ions at 6.5 nA and then thinned down using progressively lower currents until reaching a minimum of 80 pA, with the final smoothing of the lamellae utilizing 5 kV Ga ions at 80 pA. Bright-field images were recorded using a Thermo Fisher Tecnai Osiris machine operated at 200 kV accelerating voltage, with dark contrasts showing areas with strongly diffracting incident electrons.

**Calculating cogwheel interface densities and diagenesis resistance ratios.** Cogwheel interface densities were calculated using the ImageJ macro from van Dijk et al.[43]. Secondary electron SEM images at ×5000 magnification were taken of the outer surface of the second to the last chamber of tests from all three species

studied here. The raw SEM images were imported into ImageJ and the images cropped such that only areas in sharp focus were selected. Grey value thresholding was used to individually select pore spaces and cogwheel interfaces. The net surface area is the total surface area imaged minus the area of the pores. The total cogwheel interface length is the total length of all cogwheel interfaces excluding pore–calcite interfaces and pixels along the image frame. The cogwheel interface density is calculated as the ratio of the total cogwheel interface length to the net test surface area (i.e., in μm per $μm^2$). Supplementary Fig. 11 provides an overview of this process.

The diagenesis resistance ratio was calculated using a modified version of the ImageJ macro from van Dijk et al.[43] (Supplementary Software). Foraminifera tests were embedded in epoxy and polished down to the widest cross-section of the test (using diamond polishing pastes with a final grain size of 0.25 μm) and imaged using backscattered electrons (BSE) SEM. Grey value thresholding was used to select the test area to create a binary image. The paintbrush tool was used to remove extraneous pixels and to fill in fractured and broken parts of the test. The diagenesis resistance ratio is the ratio of the cross-sectional test surface area to the number of chambers, which is equal to the number of organic linings.

## Data availability

All relevant datasets for this research are included in the Supplementary Info files.

## Code availability

The ImageJ macro used in this study have been deposited in the EarthChem database accessible at https://doi.org/10.26022/IEDA/112180[72].

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

## Acknowledgements

Funding for this study was provided by the European Research Council Advanced Grant 788752. We thank Charlotte LeKieffre for providing *Ammonia* sp. and *H. germanica* and Marleen Stuhr for providing *A. lessonii* specimens. We thank Jean Daraspe, Antonio

Mucciolo, and Damien De Bellis for technical advice regarding sample preparation, Youness Zangui for technical assistance during polishing, Louise Jensen for assistance with SEM imaging, Inge van Dijk with advice on etching, and Christine Barras for helpful discussions.

## Author contributions

D.C.-L., A.A., J.G., S.B., L.P.B., D.D., J.S. and A.M. designed the experimental study; A.A. performed the isotope exchange experiments; A.A. and T.V. acquired the bulk enrichment data; D.C.-L. and A.A. acquired the SEM images; J.G., A.B. and O.G. acquired TEM data; D.C.-L. and S.E. acquired NanoSIMS data; D.C.-L., A.A. and A.M. drafted the manuscript. All co-authors contributed to the writing of the manuscript.

## Competing interests

The authors declare no competing interests.
