## [Peer Review File · Nature Communications]

Reviewers' Comments:

Reviewer #1:

None

Cisneros-Lazaro et al., Diagenetic isotope exchange of fossil foraminifera tests is fast, pervasive and species-dependent

This paper reports a study to simulate the effects of diagenesis of the $\delta^{18}\text{O}$ of foraminifera tests. It's an important topic as foraminifera $\delta^{18}\text{O}$ is used to constrain seawater $\delta^{18}\text{O}$ and seawater temperature. The study is well planned and the manuscript is well written but I did not agree that the SEM and binocular microscope images show that the tests are unaltered by the treatment. I would also like the authors to explore how the methanol/oxidative cleaning could influence the tests (compared to the simulated diagenesis treatment).

Main points:

The authors conclude that the simulated diagenesis does not visibly alter the foraminifera test. However I think several of the images do show visible alteration. Both the *H. germanica* and *Ammonia* spp specimens appear slightly more frosted after the treatment than before (Figure 1). Figure S2 also shows that the test morphology is altered by the treatment for all the foraminifera species. There is evidence for precipitation in the pore spaces in B and D (compare the images to A and C). In F the pore spaces appear smaller than in E suggesting that there has been precipitation in these pore spaces too. The observation that the simulated diagenesis causes very subtle changes in the test appearance does not invalidate the study. The key point here is do fossil foraminifera used for $\delta^{18}\text{O}$ have these subtle changes and are they still considered suitable for palaeoenvironmental analysis.

The authors have used a methanol and oxidative treatment (line 99) to remove organic materials from the test before simulated diagenesis. It would be really useful to explore how these treatments affect the test morphology and chemistry before the simulated diagenesis. So include images of methanol/oxidative treated tests in Figure 1 and Figure S2 to compare to the untreated and simulated diagenesis treatments. This would help to determine if the slight increase in the frosting of the tests in Figure 1 reflects the simulated diagenesis or the oxidative treatment. It would be interesting to include nanoSIMS images of cleaned tests to show if the cleaning is associated with any change in the $\delta^{18}\text{O}$ distribution.

The authors provide SEM images of pristine tests at high resolution (Figure S1 C-E). It would be useful to provide comparable images in tests after methanol/oxidative treatment and after simulated diagenesis to indicate if the treatments alter the test structure at this resolution.

Prior to analysis fossil foraminifera are usually subject to fragmentation and cleaning. Would this cleaning procedure remove the altered $\delta^{18}\text{O}$ sections of the foraminifera test? I would like to see a discussion of this in the manuscript.

Minor points

Line 22 Is $\delta^{18}\text{O}$ the most widely used temperature proxy? Foraminifera Mg/Ca is also common. It is fair to say that $\delta^{18}\text{O}$ is a commonly used temperature proxy.

Lines 53-56 This suggests that burial is required for diagenesis. This is not the case. Diagenesis occurs without burial.

Line 133. None of the images in Figure 2 show evidence that these interfaces are enriched in organic matter. Please rephrase this sentence.

Figure S1 does not show organic rich grain boundaries (line 141).

Lines 151-154 suggests there are no visible signs of dissolution think the images do show evidence of alteration. or precipitation. The legend for Figure S2 states negligible signs of precipitation.

Line 159-161 What are the errors? Standard deviation? SEM? Please quite then number of analyses here.

Lines 56-165. Please use a suitable statistical test to demonstrate that there is a significant difference in the d18O of the foraminifera species.

Lines 180-182. The 18O enrichment frequently does not correlate with P enrichment. E.g. Figure 4B shows that P is concentrated in thin layers which run parallel to the test surface. Figure a shows that d18O is enriched in these areas but can also be just as high in the spaces between the layers. I did not follow the conclusion that enrichment was higher at the high P areas (line 210). This distribution of enriched 18O is also observed in Figure 4G. If the P pinpoints the organic linings (line 177) then some of the 18O enrichment does not occur along these linings. Please discuss this more in the manuscript.

Line 216. Was the Iceland Spar also subject to the methanol/oxidative treatment before simulated diagenesis?

Line 230 18O enrichment often does not correlate with high P.

Line 240 Provide a reference to support the idea that Mg substitutes for Ca in calcite.

Line 245. The tests show evidence of very subtle alteration

Line 246-251 Please discuss why 18O enrichment often occur away from the P enrichment.

Lines 271-271 Figure s1 does not show organic rich areas.

Line 278 Would a P of zero be indicative of no organic matter? This would benefit from a more detailed explanation of the calibration of P in the nano SIMS. Did you measure the P of the Iceland Spar. Did this show evidence of organic matter?

Throughout the manuscript. Why do you refer to this as “early diagenesis”? It would help to explain why you term it this. I think “diagenesis” is perhaps better.

Magnitude of d18O change. Could you estimate how much the d18O of a foraminifera test would be altered by exchange with seawater?

The line numbers quoted here refer to the manuscript file where all changes have been implemented.

Response to reviewer 1:

This paper reports a study to simulate the effects of diagenesis of the d18O of foraminifera test. It's an important topic as foraminifera d18O is used to constrain seawater d18O and seawater temperature. The study is well planned and the manuscript is well written but I did not agree that the SEM and binocular microscope images show that the tests are unaltered by the treatment. I would also like the authors to explore how the methanol/oxidative cleaning could influence the tests (compared to the simulated diagenesis treatment).

Main points:

The authors conclude that the simulated diagenesis does not visibly alter the foraminifera test. However I think several of the images do show visible alteration. Both the *H. germanica* and *Ammonia* sp specimens appear slightly more frosted after the treatment than before (Figure 1). Figure S2 also shows that the test morphology is altered by the treatment for all the foraminifera species. There is evidence for precipitation in the pore spaces in B and D (compare the images to A and C). In F the pore spaces appear smaller than in E suggesting that there has been precipitation in these pore spaces too. The observation that the simulated diagenesis causes very subtle changes in the test appearance does not invalidate the study. The key point here is do fossil foraminifera used for d18O have these subtle changes and are they still considered suitable for palaeoenvironmental analysis.

In response to these constructive comments, we re-examined the surface textures of all three species using optical and scanning electron microscopy (SEM). Previously, optical images were taken on different microscopes and at different magnifications, and SEM images were taken from carbon-coated and uncoated tests. Supplementary Figs. 2 to 4 now show three images of the tests of each of three foraminifera species without cleaning, after methanol cleaning, and after oxidative treatment. Alongside these images are tests that underwent the simulated diagenesis treatment after methanol cleaning and an oxidative treatment. These images were all taken with the same microscope at a fixed magnification. Supplementary Figs. 5 to 7 show high magnification (5000×) SEM images of the surfaces of tests under the same four treatment conditions. The images were taken with a secondary electron detector on uncoated samples, at 1 kV to minimise charging. We followed recommendations in van Dijk et al.¹ to always image the second to last test chamber as the final chamber can be thin and deformed in terms of pore structure. It is clear from these images that surface pitting and 'precipitation' within the pores in the incubated foraminifera tests cannot be distinguished from the natural variability in textures present in unreacted, uncleaned samples. Therefore we are confident in stating that the textures after the simulated diagenesis treatment are not recognisably different from those seen in pristine uncleaned foraminifera tests.

The authors have used a methanol and oxidative treatment (line 99) to remove organic materials from the test before simulated diagenesis. It would be really useful to explore how these treatments affect the test morphology and chemistry before the simulated diagenesis. So include images of methanol/oxidative treated tests in Figure 1 and Figure S2 to compare to the untreated and simulated diagenesis treatments. This would help to determine if the slight increase in the frosting of the tests in Figure 1 reflects the simulated diagenesis or the oxidative treatment. It would be interesting to include nanoSIMS images of cleaned tests to show if the cleaning is associated with any change in the d18O distribution.

Supplementary Figs. 2 to 7 now include images of three different foraminifera tests in their pristine uncleaned state, after methanol only cleaning, and after methanol and oxidative cleaning to compare them with three examples of foraminifera tests that underwent the simulated diagenesis procedure. No visible difference is noted between cleaned foraminifera before and after experiments. Figure 1 has been updated to include these newly acquired images. NanoSIMS images of controls of pristine, methanol cleaned, and methanol and oxidatively cleaned tests are indistinguishable from one another and have the same $\delta^{18}\text{O}$ throughout the test within the precision of NanoSIMS measurements.

The authors provide SEM images of pristine tests at high resolution (Figure S1 C-E). It would be useful to provide comparable images in tests after methanol/oxidative treatment and after simulate diagenesis to indicative if the treatments alter the test structure at this resolution.

Yes, agreed. Thanks for pointing that out. Supplementary Fig. 1 now consists of high-resolution SEM images taken at 35 000× magnification that show the calcite nanograins in pristine tests as well as those that underwent simulated diagenesis. There were no clear differences visible even at this scale.

Prior to analysis fossil foraminifera are usually subject to fragmentation and cleaning. Would this cleaning procedure remove the altered d18O sections of the foraminifera test? I would like to see a discussion of this in the manuscript.

When the foraminifera are taken out of the gold capsules at the end of the simulated diagenesis experiment, they were rinsed with artificial seawater as well as distilled and deionized water and technical grade ethanol. As the NanoSIMS images show that the ¹⁸O-enriched fluids penetrated into the bulk of the foraminifera tests, fragmentation and cleaning would not simply ‘rinse away’ the altered isotopic composition of the tests. We have added this explanation to the text (Lines 400-402).

Minor points:

Line 22 Is d18O the most widely used temperature proxy? Foraminifera Mg/Ca is also common. It is fair to say that d18O is a commonly used temperature proxy.

Thanks- we agree.

Lines 53-56 This suggests that burial is required for diagenesis. This is not the case. Diagenesis occurs without burial.

A reference to Pearson and Burgess² (Lines 55-57) has been added to clarify that the alteration of biogenic calcite structures can start as soon as an organism dies.

Line 133. None of the images in Figure 2 show evidence that these interfaces are enriched in organic matter. Please rephrase this sentence.

The sentence has been rephrased and a reference to Cuif *et al.*³ has been added (Lines 132-134).

Figure S1 does not show organic rich grain boundaries (line 141).

The reference to Supplementary Fig. 1 has been moved and the sentence rephrased to clarify that ‘organic-rich grain boundaries’ have been demonstrated through AFM by Cuif *et al.*³ (Lines 140-142).

Lines 151-154 suggests there are no visible signs of dissolution think the images do show evidence of alteration. or precipitation. The legend for Figure S2 states negligible signs of precipitation.

The text and the figure captions have been modified to be consistent with our view that dissolution or precipitation in the experimentally diagenetically altered forams are virtually impossible to differentiate from that in pristine foraminifera. (Lines 153-155)

Line 159-161 What are the errors? Standard deviation? SEM? Please quite then number of analyses here.

The text has been modified accordingly and a table has been provided in Supplementary Table 1. (Lines 160-163)

Lines 56-165. Please use a suitable statistical test to demonstrate that there is a significant difference in the d18O of the foraminifera species.

We performed F- and T-tests to demonstrate that there is a significant difference in bulk ¹⁸O-enrichment between both *Ammonia sp.* and *Haynesina germanica* and *Ammonia sp.* and *A. lessoni* (Lines 163-167).

Lines 180-182. The ^{18}O enrichment frequently does not correlate with P enrichment. E.g. Figure 4B shows that P is concentrated in thin layers which run parallel to the test surface. Figure a shows that $\delta^{18}\text{O}$ is enriched in these areas but can also be just as high in the spaces between the layers. I did not follow the conclusion that enrichment was higher at the high P areas (line 210). This distribution of enriched ^{18}O is also observed in Figure 4G. If the P pinpoints the organic linings (line 177) then some of the ^{18}O enrichment does not occur along these linings. Please discuss this more in the manuscript.

The line scan in Fig. 5 shows that in between the organic linings, the ^{18}O -enrichment never falls below 100 ‰ above the $^{18}\text{O}/^{16}\text{O}$ ratio of an unlabelled control sample, but at the location of the organic linings it can be as high as 1500 ‰. In Figs. 4a-f, the imaging surface shows a cross-sectional view of the tests which corresponds to the front face of the schematic diagram of the cogwheel structures in Fig. 2a. The zones of increased ^{18}O -enrichment occurring as lines perpendicular to the organic linings correspond to the ‘side-view’ of the cogwheel structures. Although the ^{18}O -enrichment was highest at organic linings and cogwheel structure interfaces, it is a fundamental conclusion of this study that ^{18}O -enrichment occurred everywhere in the tests to some extent. This is thoroughly explained in the Discussion section titled “Pervasive ^{18}O -enrichments” (Lines 270-293).

Line 216. Was the Iceland Spar also subject to the methanol/oxidative treatment before simulated diagenesis?

The Iceland Spar was not subjected to methanol/oxidative cleaning because fluorescence imaging did not show the presence of any organic compounds, nor were there any traces of any other surficial mineral contaminants. The text now clarifies that the Iceland spar was not cleaned (Line 433).

Line 230 ^{18}O enrichment often does not correlate with high P.

See comment above.

Line 240 Provide a reference to support the idea that Mg substitutes for Ca in calcite.

A reference to Branson *et al.*⁴ which shows that Mg substitutes ideally for Ca has been added (Line 245).

Line 245. The tests show evidence of very subtle alteration

See Supplementary Figs. 2 to 7 and comments above.

Line 246-251 Please discuss why ^{18}O enrichment often occur away from the P enrichment.

See comment above.

Lines 271-271 Figure s1 does not show organic rich areas.

The reference has been moved to make it clear that Supplementary Fig. 1 only shows the calcite particles (Lines 278-280).

Line 278 Would a P of zero be indicative of no organic matter? This would benefit from a more detailed explanation of the calibration of P in the nano SIMS. Did you measure the P of the Iceland Spar. Did this show evidence of organic matter?

P signals of zero would not necessarily be indicative of no organic matter, but a P signal has been used here, as in Geerken *et al.*⁵, to positively identify organic molecules. There was no P signal in the Iceland spar.

Throughout the manuscript. Why do you refer to this as “early diagenesis”? It would help to explain why you term it this. I think “diagenesis” is perhaps better.

We agree - diagenesis is indeed better.

Magnitude of $\delta^{18}\text{O}$ change. Could you estimate how much the $\delta^{18}\text{O}$ of a foraminifera test would be altered by exchange with seawater?

This question, which is very important, was investigated (modelled) in detail by Bernard *et al.*⁶ who showed that, even if driven by the slowest possible process, i.e., solid state diffusion, isotopic exchange with seawater can significantly change the bulk isotopic composition of a foraminifera test buried in an ocean sediment on geological timescales. Our work directly visualizes the pathways of

isotopic exchange and thereby adds a new level of sophistication to the understanding of how oxygen isotopic exchange proceeds in biogenic calcites.

Response to reviewer 2:

Authors Cisneros-Lazaro et al. submitted a manuscript titled, “Diagenetic isotope exchange of fossil foraminifera tests is fast, pervasive, and species-dependent”. The authors collected recent marine forams from mudflats and separated the samples into three species. The authors cleaned the forams using known oxidative cleaning procedures. They then exposed the forams and an Iceland spar calcite to $\delta^{18}\text{O}$ enriched water at high temperature for varying lengths of time. All species of forams became enriched in $\delta^{18}\text{O}$ values and nanoSIM work showed that areas where organic matter were present served as pathways for the enriched fluid. The spar calcite did not change. The authors have two major conclusions: 1) that all geologic aged forams were likely altered and 2) that the glassy texture is not a perfect indicator of the preservation state of forams.

Overall, the manuscript provides a robust and important investigation into the using forams for paleoclimate/environmental reconstructions. The manuscript is clear and generally well written. The manuscript does an excellent job of demonstrating that forams considered ‘pristine’ based on visual inspection could actually be altered based. This is a strong and important conclusion. However, the manuscript is lacking supportive documentation that all geologic aged forams were likely to have been altered. A more robust evaluation will only strengthen the argument or provide more accurate language to represent the conclusions. Two major concerns with this conclusion are outlined below:

1. The authors assume that early diagenesis is with a fluid substantially different than that in which the forams formed. What if there is no fluid or the fluid is similar to the formation fluid? How does time play a factor? The manuscript does not provide results between the 4-hour vs. 6-day experiments. Taking the results and producing a rate equation for exchange that can be used on geologic timescales would be a nice addition to the paper.

Under the *Paleo-environmental reconstructions based on biogenic calcites- an outlook* section (Lines 364-372), we now explain that all fossil forams from ocean sediments used for paleo-environmental reconstruction are surrounded by porefluids because these sediments, even when clay-rich, remain porous (30–40 % porosity) up to 1000 m burial depth. In addition, even foraminifera hosted in clay-rich sediments surrounded by fluids with the exact same initial isotopic composition as that in which the foraminifera lived, will isotopically re-equilibrate due to the isotopic disequilibrium which results from the increased temperatures during burial. The 4 hour and 6 day long experiments that we conducted here do not allow us to properly extrapolate to geologic timescales, but we agree with the reviewer that this is something that should be done. We provided a model of the impact of isotopic exchange over geological timescales in a previous study [Bernard *et al.* ⁶]).

2. The authors assume that all forams experience oxidative cleaning during diagenesis, however the manuscript does not state at what temperature or how long a foram needs to be exposed to those conditions to substantially change the isotopic composition. It is still standard to clean fossil forams; therefore, some organic matter must remain. Is the removal even necessary? Did the authors conduct the experiment on uncleaned forams? This may be important since the authors claim that much of the exchange would have happened during early diagenesis when the organic matter is being removed, yet the experiments were run after most of the organic matter was removed.

We have performed experiments on uncleaned foraminifera; however, these experiments led to experimental challenges. Uncleaned foraminifera are filled with and covered by a multitude of clay minerals, silicate grains, and carbonate grains from other organisms, e.g., foraminifera remains, diatoms, pelloids etc. ... The presence of multiple mineral species makes it impossible to maintain chemical equilibrium during experimental time frames without knowing the chemical composition of the uncleaned foraminifera aliquot. Additionally, the large amount of clay minerals that can adhere to some uncleaned foraminifera tests can act as a sink for ^{18}O , which can change the oxygen isotopic composition of the fluid during the experimental time frame. To maintain experimental

conditions with a consistent set of chemical and isotopic compositions, all foraminifera tests had to be cleaned. To clarify, our results indicate that isotopic exchange happens anywhere fluids penetrate into the spaces created by the removal of the organics. This can happen any time the foraminifera tests come into contact with fluids of a different isotopic composition or temperature, which is not only during early diagenesis when the organic matter is being removed.

Lastly, there is a lack of transparency in the data. Perhaps a results table in the supplementary file would be appropriate.

Done. The individual measurements for the bulk oxygen isotope measurements are provided in Supplementary Table 1, the new cogwheel interface density quantification parameters are provided in Supplementary Table 2, and the new parameters used to calculate test thicknesses and the diagenesis resistance factor for *Ammonia* and *H. germanica* are provided in Supplementary Table 3.

This manuscript is an important contribution to scientists looking using forams for paleoclimate reconstruction and for the broader paleoclimate/climate modelling community. The topic of the manuscript is fit for Nature Communications. The reviewer suggests a relatively major revision before publication. Once either the language regarding the fossil forams is changed or elaborated on and the major concerns are addressed, this manuscript would be acceptable for publication. This reviewer is willing to re-review if necessary.

Minor line by line comments are outlined below.

Title: To the reviewer, this title is a little misleading. The authors did not actually perform these tests on fossil forams. "Fast" would be limited to the decay of organic matter which really depends on burial setting.

The word fossil has been removed. We maintain that the isotopic exchange which this paper describes is indeed fast, as the results show that isotopic exchange occurs within the bulk of the calcite within hours to days. On geological timescales this isotopic exchange is instantaneous.

Line 33-34: Not true, please re-word change 'likely to' to 'may'. Just because it is possible does not mean that it happened. The tests would need to be exposed to a different fluid at a different temperature in order to actually exchange.

Done.

Line 67: Although the studies are later listed, may be good to also cite them after "recent studies".

Done.

Line 93: Initially 'pristine' but the purpose of the paper to suggest none are pristine. Maybe reword? Perhaps add the word 'visually'.

The tests used in this study were from recently living modern visually-pristine foraminifera. We have added the word visually to the test descriptions.

Line 307: density of cogwheel interfaces. Is this quantitative or subjective? Does the manuscript describe how the authors defined the density of cogwheels?

In response to this comment, and following the method described in van Dijk *et al.*¹, the density of cogwheel structures of 3 foraminifera tests from each of the 3 species have now been quantified. Cogwheel interface density (μm per μm^2) has been quantified as the ratio of the total cogwheel interface length (the sum of all cogwheel interfaces) to the net test surface area, which is the total surface area minus the pores. This has been added to the manuscript in Lines 315-323.

Line 318: "in" should be changed to "than", currently as written the sentence makes it seem like the $\delta^{18}\text{O}$ enrichment is largest in *Ammonia* sp.

Done.

Line 307-318: Overall, this paragraph should be expanded to include more quantitative evidence to back up the author's hypothesis. Perhaps try to give a cogwheel density parameter, normalized to size. Just looking at the one example of each species in Fig. 2 did not make the density of cogwheels obvious because the magnification is different for each picture (scale bars are not all equal). How variable is the cogwheel density within one species? An equation to show the difference rate of exchange per species. Etc.

We fully agree and this comment inspired us to add new work along these lines. Fig. 2 has been modified to show images with an equal magnification for each example of the three species. Supplementary Fig. 11 and Supplementary Table 2 explain in detail how the cogwheel interface density parameter was calculated following the method in van Dijk *et al.*¹. The average cogwheel interface density ($\mu\text{m per } \mu\text{m}^2$) $\pm 1\sigma$ for *Ammonia sp.* is 1.80 ± 0.36 , 1.78 ± 0.10 for *H. germanica* and 3.52 ± 0.18 for *A. lessonii* (Lines 318-321). The higher variability for *Ammonia sp.* is likely due to the foraminifera tests used in this study being a mixture of phylotypes T1 and T6. Additionally, the thickness differences between *Ammonia sp.* and *H. germanica* have been estimated (Lines 331-344).

Lines 322-332: As stated in the general comments, the reviewer is not convinced the manuscript fully supported this large claim that all forams must have experienced extensive change. If the fluid during early diagenesis is similar to the fluid that the foram grew in, how would exchange happen? The $\delta^{18}\text{O}$ value would not change. What if the F/R ratio is low or no fluid is present during diagenesis?

See comment above.

Figure 3: Perhaps put a y-axis break and start at 250%? There is a lot of unnecessary white space in the figure.

Done.

References

1. van Dijk, I. Van, Raitzsch, M., Brummer, G. A., Bijma, J. & Goleñ, J. P. Novel Method to Image and Quantify Cogwheel Structures in Foraminiferal Shells. *Front. Ecol. Evol.* **8**, 1–13 (2020).
2. Pearson, P. N. & Burgess, C. E. Foraminifer test preservation and diagenesis: comparison of high latitude Eocene sites. *Geol. Soc. London, Spec. Publ.* **303**, 59–72 (2008).
3. Cuif, J. P., Dauphin, Y. & Sorauf, J. E. *Bioinorganics and fossils through time*. (Cambridge University Press, 2010).
4. Branson, O. *et al.* The coordination of Mg in foraminiferal calcite. *Earth Planet. Sci. Lett.* **383**, 134–141 (2013).
5. Geerken, E. *et al.* Element banding and organic linings within chamber walls of two benthic foraminifera. *Sci. Rep.* **9**, 1–15 (2019).
6. Bernard, S., Daval, D., Ackerer, P., Pont, S. & Meibom, A. Burial-induced oxygen-isotope re-equilibration of fossil foraminifera explains ocean paleotemperature paradoxes. *Nat. Commun.* **8**, 1–10 (2017).

Reviewers' Comments:

Reviewer #1:

Remarks to the Author:

This is my second review of this manuscript after the authors revised the paper on the basis of comments by me and another reviewer. The authors have done a good job on addressing most of the comments we made and I think the paper could be accepted in its current form.

In my previous review I commented that I thought some of the images of tests which had been through the simulated diagenesis procedure displayed evidence of alteration. I did not keep a copy of the original paper but I think these images have now been removed from the paper. The current manuscript states that all figures were collected on the same microscope, same magnification, same light etc to facilitate comparison. It may be that the previous images were removed due to discrepancies in image collection but if not then the authors could consider keeping these original images in the published paper e.g. in supplementary data. The fact that some of the forams show evidence of alteration does not negate the paper's main findings but perhaps gives a more holistic overview of the whole dataset.

Reviewer #2:

Remarks to the Author:

The authors incorporated all reviewer comments sufficiently. The manuscript is recommended to be accepted.

Response to Reviewers

Reviewer #1 (Remarks to the Author):

This is my second review of this manuscript after the authors revised the paper on the basis of comments by me and another reviewer. The authors have done a good job on addressing most of the comments we made and I think the paper could be accepted in its current form.

In my previous review I commented that I thought some of the images of tests which had been through the simulated diagenesis procedure displayed evidence of alteration. I did not keep a copy of the original paper but I think these images have now been removed from the paper. The current manuscript states that all figures were collected on the same microscope, same magnification, same light etc to facilitate comparison. It may be that the previous images were removed due to discrepancies in image collection but if not then the authors could consider keeping these original images in the published paper e.g. in supplementary data. The fact that some of the forams show evidence of alteration does not negate the papers main findings but perhaps gives a more holistic overview of the whole dataset.

We thank the reviewer for the careful attention dedicated to our manuscript.

As we stated in our previous response to the reviews, we replaced all the previous optical microscopy images with images strictly under the same conditions (i.e., magnification and camera settings etc.) The first version of our manuscript contained only one example of each species before and after the simulated diagenesis treatment. In the modified version of the manuscript we now document three examples of each foraminifera species before simulated diagenesis treatment, after each experimental cleaning step, as well as after the simulated diagenesis treatment (Supplementary Figs. 2-4). In addition, high magnification SEM images of the surfaces of foraminifera tests before and after the simulated diagenesis treatment (Supplementary Figs. 5-7) are now included in the manuscript to clearly show that, even at this level of observation, it is impossible to distinguish foraminifera tests that have been experimentally altered from those that have not. This is now a complete package of documentation that provides the best possible overview of the foraminifera tests from our experiments.

Reviewer #2 (Remarks to the Author):

The authors incorporated all reviewer comments sufficiently. The manuscript is recommended to be accepted.

Thank you for your positive appraisal of our resubmitted manuscript.